# Model-Based Control with Sparse Neural Dynamics

**Ziang Liu**[1,2]    **Genggeng Zhou**[2*]   **Jeff He**[2*]    **Tobia Marcucci**[3]
**Li Fei-Fei**[2]    **Jiajun Wu**[2]    **Yunzhu Li**[2,4]
[1]Cornell University    [2]Stanford University
[3]Massachusetts Institute of Technology
[4]University of Illinois Urbana-Champaign
`ziangliu@cs.cornell.edu`
`{g9zhou,jeff2024}@stanford.edu`
`tobiam@mit.edu`
`{feifeili,jiajunwu}@cs.stanford.edu`
`yunzhuli@illinois.edu`

## Abstract

Learning predictive models from observations using deep neural networks (DNNs) is a promising new approach to many real-world planning and control problems. However, common DNNs are too unstructured for effective planning, and current control methods typically rely on extensive sampling or local gradient descent. In this paper, we propose a new framework for integrated model learning and predictive control that is amenable to efficient optimization algorithms. Specifically, we start with a ReLU neural model of the system dynamics and, with minimal losses in prediction accuracy, we gradually sparsify it by removing redundant neurons. This discrete sparsification process is approximated as a continuous problem, enabling an end-to-end optimization of both the model architecture and the weight parameters. The sparsified model is subsequently used by a mixed-integer predictive controller, which represents the neuron activations as binary variables and employs efficient branch-and-bound algorithms. Our framework is applicable to a wide variety of DNNs, from simple multilayer perceptrons to complex graph neural dynamics. It can efficiently handle tasks involving complicated contact dynamics, such as object pushing, compositional object sorting, and manipulation of deformable objects. Numerical and hardware experiments show that, despite the aggressive sparsification, our framework can deliver better closed-loop performance than existing state-of-the-art methods. [†]

## 1   Introduction

Our mental model of the physical environment enables us to easily carry out a broad spectrum of complex control tasks, many of which lie far beyond the capabilities of present-day robots [32]. It is, therefore, desirable to build predictive models of the environment from observations and develop optimization algorithms to help the robots understand the impact of their actions and make effective plans to achieve a given goal. Physics-based models [26, 73] have excellent generalization ability but typically require full-state information of the environment, which is hard and sometimes impossible to obtain in complicated robotic (manipulation) tasks. Learning-based dynamics modeling circumvents the problem by learning a predictive model directly from raw sensory observations, and recent successes are rooted in the use of deep neural networks (DNNs) as the functional class [14, 21, 56, 47]. Despite their improved prediction accuracy, DNNs are highly nonlinear,

---

[*]denotes equal contribution

[†]Please see our website at robopil.github.io/Sparse-Dynamics/ for additional visualizations.

37th Conference on Neural Information Processing Systems (NeurIPS 2023).

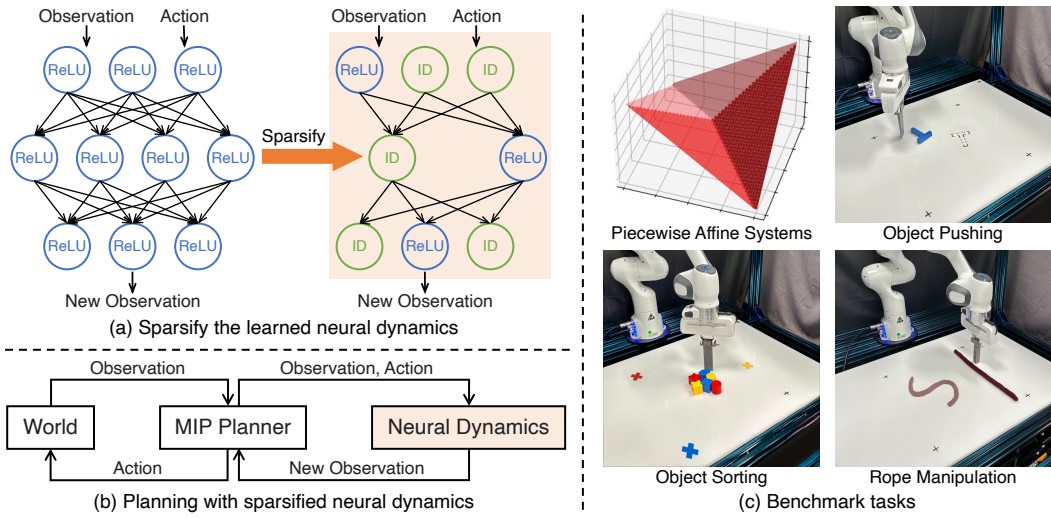

Figure 1: **Model-based control with sparse neural dynamics.** (a) Our framework sparsifies the neural dynamics models by either removing neurons or replacing ReLU activation functions with identity mappings (ID). (b) The sparsified models enable the use of efficient MIP methods for planning, which can achieve better closed-loop performance than sampling-based alternatives commonly used in model-based RL. (c) We evaluate our framework on various dynamical systems that involve complex contact dynamics, including tasks like object pushing and sorting, and manipulating a deformable rope.

making model-based planning with neural dynamics models very challenging. Existing methods often rely on extensive sampling or local gradient descent to compute control signals, and can be ineffective for complicated and long-horizon planning tasks.

Compared to DNNs, simpler models like linear models are amenable to optimization tools with better guarantees, but often struggle to accurately fit observation data. An important question arises: how precise do these models need to be when employed within a feedback control loop? The cognitive science community offers substantial evidence suggesting that humans do not maintain highly accurate mental models; nevertheless, these less precise models can be effectively used with environmental feedback [30, 10]. This notion is also key in control-oriented system identification [25, 44] and model order reduction [55, 61]. The framework from Li et al. [38] trades model expressiveness and precision for more efficient and effective optimization-based planning through the learning of compositional Koopman operators. However, their approach is limited by the linearity of the representation in the Koopman embedding space and struggles with more complex dynamics.

In this paper, we propose a framework for integrated model learning and control that trades off prediction accuracy for the use of principled optimization tools. Drawing inspiration from the neural network pruning and neural architecture search communities [22, 54, 16, 5, 41], we start from a neural network with ReLU activation functions and gradually reduce the nonlinearity of the model by removing ReLU units or replacing them with identity mappings (Figure 1a). This yields a highly sparsified neural dynamics model, that is amenable to model-based control using state-of-the-art solvers for mixed-integer programming (MIP) (Figure 1b).

We present examples where the proposed sparsification pipeline can determine region partition and uncover the underlying system for simple piecewise affine systems. Moreover, it can maintain high prediction accuracy for more complex manipulation tasks, using a considerably smaller portion of the original nonlinearities. Importantly, our approach allows the joint optimization of the network architecture and weight parameters. This yields a spectrum of models with varying degrees of sparsification. Within this spectrum, we can identify the simplest model that is adequate to meet the requirements of the downstream closed-loop control task.

Our contributions can be summarized as follows: (i) We propose a novel formulation for identifying the dynamics model from observation data. For this step, we introduce a continuous approximation of the sparsification problem, enabling end-to-end gradient-based optimization for both the model class and the model parameters (Figure 1a). (ii) By having significantly fewer ReLU units than

the full model, the sparsified dynamics model allows us to solve the predictive-control problems using efficient MIP solvers (Figure 1b). This can lead to better closed-loop performance compared to both model-free and model-based reinforcement learning (RL) baselines. (iii) Our framework can be applied to many types of neural dynamics, from vanilla multilayer perceptrons (MLPs) to complex graph neural networks (GNNs). We show its effectiveness in a variety of simulated and real-world manipulation tasks with complex contact dynamics, such as object pushing and sorting, and manipulation of deformable objects (Figure 1c).

## 2   Related Work

**Model learning for planning and control.** Model-based RL agents learn predictive models of their environment from observations, which are subsequently used to plan their actions [9, 53]. Recent successes in this domain often heavily rely on DNNs, exhibiting remarkable planning and control results in challenging simulated tasks [58], as well as complex real-world locomotion and manipulation tasks [34, 56]. Many of these studies draw inspiration from advancements in computer vision, learning dynamics models directly in pixel-space [15, 11, 12, 71, 62], keypoint representation [31, 47, 39], particle/mesh representation [36, 60, 27], or low-dimensional latent space [65, 1, 21, 20, 58, 69]. While previous works typically assume that the model class is given and fixed during the optimization process, our work puts emphasis on finding the desired model class via an aggressive network sparsification, to support optimization tools with better guarantees. We are willing to sacrifice the prediction accuracy for better closed-loop performance using more principled optimization techniques.

**Network sparsification.** The concept of neural network sparsification is not new and traces back to the 1990s [33]. Since then, extensive research has been conducted, falling broadly into two categories: network pruning [23, 22, 66, 54, 35, 24, 3, 43, 16, 42, 5, 72] and neural architecture search [74, 8, 40, 13, 64]. Many of these studies have demonstrated that fitting an overparameterized model before pruning yields better results than directly fitting a smaller model. Our formulation is closely related to DARTS [41] and FBNet [68], which both seek a continuous approximation of the discrete search process. However, unlike typical structured network compression methods, which try to remove as many units as possible, our goal here is to minimize the model nonlinearity. To this end, our method also permits the substitution of ReLU activations with identity mappings. This leaves the number of units unchanged but makes the downstream optimization problem much simpler.

**Mixed-integer modeling of neural networks.** The input-output map of a neural network with ReLU activations is a piecewise affine function that can be modeled exactly through a set of mixed-integer linear inequalities. This allows us to use highly-effective MIP solvers for the solution of the model-based control problem. The same observation has been leveraged before for robustness analysis of DNNs in [63, 70], while the efficiency of these mixed-integer models has been thoroughly studied in [2].

## 3   Method

In this section, we describe our methods for learning a dynamics model using environmental observations and for sparsifying DNNs through a continuous approximation of the discrete pruning process. Then we discuss how the sparsified model can be used by an MIP solver for trajectory optimization and closed-loop control.

### 3.1   Learning a dynamics model over the observation space

Assume we have a dataset $\mathcal{D} = \{(y_t^m, u_t^m) \mid t = 1, \ldots, T, m = 1, \ldots, M\}$ collected via interactions with the environment, where $y_t^m$ and $u_t^m$ denote the observation and action obtained at time $t$ in trajectory $m$. Our goal is to learn an autoregressive model $\hat{f}_\theta$, parameterized by $\theta$, as a proxy of the real dynamics that takes a small sequence of observations and actions from time $t'$ to the current time $t$, and predicts the next observation at time $t + 1$:

$$\hat{y}_{t+1}^m = \hat{f}_\theta(y_{t':t}^m, u_{t':t}^m). \tag{1}$$

We optimize the parameter $\theta$ to minimize the simulation error that describes the long-term discrepancy between the prediction and the actual observation:

$$\mathcal{L}(\theta) = \sum_{m=1}^{M} \sum_{t} \|y_{t+1}^m - \hat{f}_\theta(\hat{y}_{t':t}^m, u_{t':t}^m)\|_2^2. \tag{2}$$

## 3.2 Neural network sparsification by removing or replacing ReLU activations

We instantiate the transition function $\hat{f}_\theta$ as a ReLU neural network with $N$ hidden layers. Let us denote the number of neurons in the $i^{\text{th}}$ layer as $N_i$. When given an input $x = (y_{t':t}^m, u_{t':t}^m)$, we denote the value of the $j^{\text{th}}$ neuron in layer $i$ before the ReLU activation as $x_{ij}$. Regular ReLU neural networks apply the rectifier function to every $x_{ij}$ and obtain the activation value using $x_{ij}^+ = h_{ij}(x_{ij}) \equiv \text{ReLU}(x_{ij}) \triangleq \max(0, x_{ij})$. The nonlinearity introduced by the ReLU function allows the neural networks to fit the dataset but makes the downstream planning and control tasks more challenging. As suggested by many prior works in the field of neural network compression [22, 16], a lot of these ReLUs are redundant and can be removed with minimal effects on the prediction accuracy. In this work, we reduce the number of ReLU functions by replacing the function $h_{ij}$ with either an identity mapping $\text{ID}(x_{ij}) \triangleq x_{ij}$ or a zero function $\text{Zero}(x_{ij}) \triangleq 0$, where the latter is equivalent to removing the neuron (Figure 1a).

We divide the parameters in $\hat{f}_\theta$ into two vectors, $\theta = (\omega, \alpha)$. The vector $\omega$ collects the weight matrices and the bias terms. The vector $\alpha$ consists of a set of integer variables that parameterize the architecture of the neural network: $\alpha = \{\alpha_{ij} \in \{1, 2, 3\} \mid i = 1, \ldots, N, j = 1, \ldots, N_i\}$, such that

$$h_{ij}(x_{ij}) = \begin{cases} \text{ReLU}(x_{ij}) & \text{if } \alpha_{ij} = 1 \\ \text{ID}(x_{ij}) & \text{if } \alpha_{ij} = 2 \\ \text{Zero}(x_{ij}) & \text{if } \alpha_{ij} = 3 \end{cases}. \tag{3}$$

The sparsification problem can then be formulated as the following MIP:

$$\min_{\theta = (\omega, \alpha)} \quad \mathcal{L}(\theta) \qquad \text{s.t.} \quad \sum_{i=1}^{N} \sum_{j=1}^{N_i} \mathbb{1}(\alpha_{ij} = 1) \leq \varepsilon, \tag{4}$$

where $\mathbb{1}$ is the indicator function, and the value of $\varepsilon$ decides the number of regular ReLU functions that are allowed to remain in the neural network.

## 3.3 Reparameterizing the categorical distribution using Gumbel-Softmax

Solving the optimization problem in Equation 4 is hard, as the number of integer variables in $\alpha$ equals the number of ReLU neurons in the neural network, which is typically very large. Therefore, we relax the problem by introducing a random variable $\pi_{ij}$ indicating the categorical distribution of $\alpha_{ij}$ assigning to one of the three categories, where $\pi_{ij}^k \triangleq p(\alpha_{ij} = k)$ for $k = 1, 2, 3$. We can then reformulate the problem as:

$$\min_{\omega, \pi} \quad \mathbb{E}[\mathcal{L}(\theta)] \qquad \text{s.t.} \quad \sum_{i=1}^{N} \sum_{j=1}^{N_i} \pi_{ij}^1 \leq \varepsilon, \quad \alpha_{ij} \sim \pi_{ij}, \tag{5}$$

where $\pi \triangleq \{\pi_{ij} \mid i = 1, \ldots, N, j = 1, \ldots, N_i\}$.

In Equation 5, the sampling procedure $\alpha_{ij} \sim \pi_{ij}$ is not differentiable. In order to make end-to-end gradient-based optimization possible, we employ the Gumbel-Softmax [28, 46] technique to obtain a continuous approximation of the discrete distribution.

Specifically, for a 3-class categorical distribution $\pi_{ij}$, where the class probabilities are denoted as $\pi_{ij}^1, \pi_{ij}^2, \pi_{ij}^3$, Gumbel-Max [17] allows us to draw 3-dimensional one-hot categorical samples $\hat{z}_{ij}$ from the distribution via:

$$\hat{z}_{ij} = \text{OneHot}(\arg\max_k(\log \pi_{ij}^k + g^k)), \tag{6}$$

where $g^k$ are i.i.d. samples drawn from Gumbel$(0, 1)$, which is obtained by sampling $u^k \sim$ Uniform$(0, 1)$ and computing $g^k = -\log(-\log(u^k))$. We can then use the softmax function as a continuous, differentiable approximation of the $\arg\max$ function:

$$z_{ij}^k = \frac{\exp\left((\log \pi_{ij}^k + g^k)/\tau\right)}{\sum_{k'} \exp\left((\log \pi_{ij}^{k'} + g^{k'})/\tau\right)}. \tag{7}$$

We denote this operation as $z_{ij} \sim \text{Concrete}(\pi_{ij}, \tau)$ [46], where $\tau$ is a temperature parameter controlling how close the softmax approximation is to the discrete distribution. As the temperature $\tau$ approaches zero, samples from the Gumbel-Softmax distribution become one-hot and identical to the original categorical distribution.

After obtaining $z_{ij}$, we can calculate the activation value $x_{ij}^+$ as a weighted sum of different functional choices:

$$x_{ij}^+ = \hat{h}_{ij}(x_{ij}) \triangleq z_{ij}^1 \cdot \text{ReLU}(x_{ij}) + z_{ij}^2 \cdot \text{ID}(x_{ij}) + z_{ij}^3 \cdot \text{Zero}(x_{ij}), \tag{8}$$

and then use gradient descent to optimize both the weight parameters $\omega$ and the architecture distribution parameters $\pi$.

During training, we can also constrain $z_{ij}$ to be one-hot vectors by using $\arg\max$, but use the continuous approximation in the backward pass by approximating $\nabla_\theta \hat{z}_{ij} \approx \nabla_\theta z_{ij}$. This is denoted as "Straight-Through" Gumbel Estimator in [28].

## 3.4 Optimization algorithm

Instead of limiting the number of regular ReLUs from the very beginning of the training process, we start with a randomly initialized neural network and use gradient descent to optimize $\omega$ and $\pi$ by minimizing the following objective function until convergence:

$$\mathbb{E}[\mathcal{L}(\theta)] + \lambda R(\pi), \tag{9}$$

where the regularization term $R(\pi) \triangleq \sum_{i=1}^N \sum_{j=1}^{N_i} \pi_{ij}^1$ aims to explicitly reduce the use of the regular ReLU function. One could also consider adjusting it to $R(\pi) \triangleq \sum_{i=1}^N \sum_{j=1}^{N_i} (\pi_{ij}^1 + \lambda_{\text{ID}} \pi_{ij}^2)$ with a small $\lambda_{\text{ID}}$ to discourage the use of identity mappings at the same time.

We then take an iterative approach by starting with a relatively large $\varepsilon_1$ and gradually decrease its value for $K$ iterations with $\varepsilon_1 > \varepsilon_2 > \cdots > \varepsilon_K = \varepsilon$. Within each optimization iteration using $\varepsilon_k$, we first rank the neurons according to $\max(\pi_{ij}^2, \pi_{ij}^3)$ in descending order, and assign the activation function for the top-ranked neurons as ID if $\pi_{ij}^2 \geq \pi_{ij}^3$, or Zero otherwise, while keeping the bottom $\varepsilon_k$ neurons intact using Gumbel-Softmax as described in Section 3.3. Subsequently, we continue optimizing $\omega$ and $\pi$ using gradient descent to minimize Equation 9. The sparsification process generates a range of models at various sparsification levels for subsequent investigations.

## 3.5 Closed-loop feedback control using the sparsified models

After we have obtained the sparsified dynamics models, we fix the model architecture and formulate the model-based planning task as the following trajectory optimization problem:

$$\min_u \quad \sum_t c(y_t, u_t) \qquad \text{s.t.} \quad y_{t+1} = \hat{f}_\theta(y_{t':t}, u_{t':t}), \tag{10}$$

where $c$ is the cost function. When the transition function $\hat{f}_\theta$ is a highly nonlinear neural network, solving the optimization problem is not easy. Previous methods [71, 11, 56, 14, 47] typically regard the transition function as a black box and rely on sampling-based algorithms like the cross-entropy method (CEM) [57] and model-predictive path integral (MPPI) [67] for online planning. Others have also tried applying gradient descent to derive the action signals [36, 37]. However, the number of required samples grows exponentially with the number of inputs and trajectory length. Gradient descent can also be stuck in local optima, and it is also hard to assess the optimality or robustness of the derived action sequence using these methods.

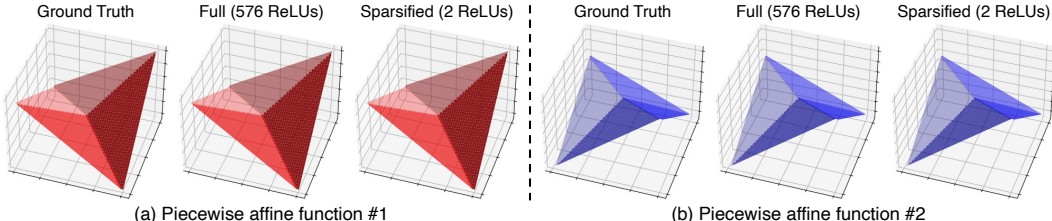

| Ground Truth | Full (576 ReLUs) | Sparsified (2 ReLUs) | Ground Truth | Full (576 ReLUs) | Sparsified (2 ReLUs) |

(a) Piecewise affine function #1      (b) Piecewise affine function #2

Figure 2: **Recover the ground truth piecewise affine functions from data.** We evaluate our sparsification pipeline on two hand-designed piecewise affine functions composed of four linear pieces. Our pipeline successfully generates sparsified models with 2 ReLUs that accurately fit the data, determine the region partition, and recover the underlying ground truth system.

### 3.5.1 Mixed-integer formulation of ReLU neural dynamics

The sparsified neural dynamics models open up the possibility of dissecting the model and solving the problem using more principled optimization tools. Specifically, given that a ReLU neural network is a piecewise affine function, we can formulate Equation 10 as MIP. We assign to each ReLU a binary variable $a = \mathbb{1}(x \geq 0)$ to indicate whether the pre-activation value is larger or smaller than zero. Given lower and upper bounds on the input $l \leq x \leq u$ (which we calculate by passing the offline dataset through the sparsified neural networks), the equality $x^+ = \text{ReLU}(x) \triangleq \max(0, x)$ can be modeled through the following set of mixed-integer linear constraints:

$$x^+ \leq x - l(1 - a), \quad x^+ \geq x, \quad x^+ \leq ua, \quad x^+ \geq 0, \quad a \in \{0, 1\}. \tag{11}$$

If only a few ReLUs are left in the model, Equation 10 can be efficiently solved to global optimality.

The formulation in Equation 11 is the simplest mixed-integer encoding of a ReLU network, and a variety of strategies are available in the literature to accelerate the solution of our MIPs. For large-scale models, it is possible to *warm start* the optimization process using sampling-based methods or gradient descent, and subsequently refine the solution using MIP solvers [49]. There also exist more advanced techniques to formulate the MIP [2, 48, 50], these can lead to tighter convex relaxations of our problem and allow us to identify high-quality solutions of Equation 10 earlier in the branch-and-bound process. The ability to find globally-optimal solutions is attractive but requires the model to exhibit a reasonable global performance. The sparsification step helps us also in this direction, since we typically expect a smaller simulation error from a sparsified (simpler) model than its unsparsified (very complicated) counterpart when moving away from the training distribution. In addition, we could also explicitly counteract this issue with the addition of trust-region constraints that prevent the optimizer from exploiting model inaccuracies in the areas of the input space that are not well-supported by the training data [52].

### 3.5.2 Tradeoff between model accuracy and closed-loop control performance

Models with fewer ReLUs are generally less accurate but permit the use of more advanced optimization tools, like efficient branch-and-bound algorithms implemented in state-of-the-art solvers. Within a model-predictive control (MPC) framework, the controller can leverage the environmental feedback to counteract prediction errors via online modifications of the action sequence. The iterative optimization procedure in Section 3.4 yields a series of models at different sparsification levels. By comparing their performances and investigating the trade-off between prediction accuracy and closed-loop control performance, we can select the model with the most desirable capacity.

## 4 Experiments

In our experiments, we seek to address three central questions: (1) How does the varying number of ReLUs affect the prediction accuracy? (2) How does the varying number of ReLUs affect open-loop planning? (3) Can the sparsified model, when combined with more principled optimization methods, deliver better closed-loop control results?

**Environments, tasks, and model classes.** We evaluate our framework on four environments specified in different observation spaces, including state, keypoints, and object-centric representations.

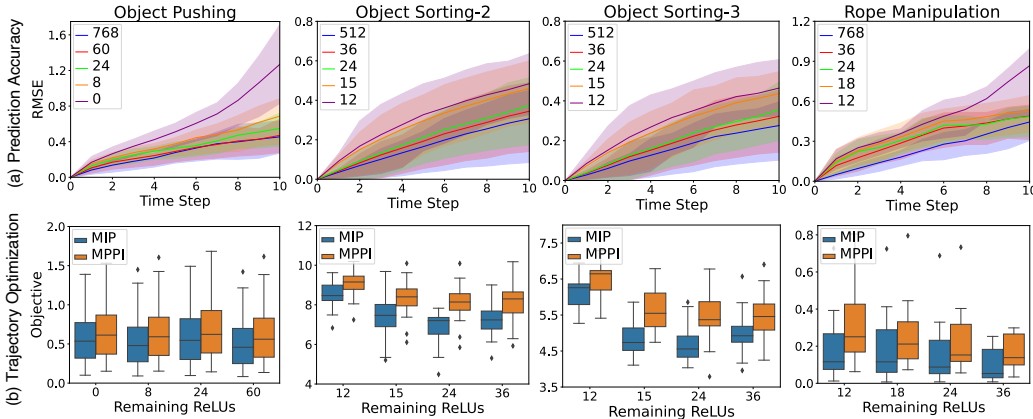

Figure 3: **Quantitative analysis of the sparsified models for open-loop prediction and planning.** (a) Long-term future prediction error, with the shaded area representing the region between the 25th and 75th percentiles. The significant overlap between the curves suggests that reducing the number of ReLUs only leads to a minimal decrease in prediction accuracy. (b) Results of the trajectory optimization problem from Equation 10. We compare two optimization formulations: mixed-integer programming (MIP) and model-predictive path integral (MPPI), using models with varying levels of sparsification. The figure clearly indicates that MIP consistently outperforms its sampling-based counterpart, MPPI.

These evaluation environments make use of different modeling classes, including vanilla MLPs and complex GNNs. For closed-loop control evaluation, we additionally present the performance of our framework on two standardized benchmark environments from OpenAI Gym [7], `CartPole-v1` and `Reacher-v4`.

- **Piecewise affine function.** We consider manually designed two-dimensional piecewise affine functions consisting of four linear pieces (Figure 2), and the goal is to recover the ground-truth system from data through the sparsification process starting from an overparameterized MLP. To train the model, we collect 1,600 transition pairs from the ground truth functions uniformly distributed over the 2D input space.

- **Object pushing.** A fully-actuated pusher interacts with an object moving on a 2D plane, as depicted in Figure 4a. The goal is to manipulate the object to reach a randomly generated target pose. We generated 50,000 transition pairs using the Pymunk simulator [6]. The observation $y_t$ is defined by the position of four selected keypoints on the object, and the dynamics model is also instantiated as an MLP.

- **Object sorting.** In Figure 4c, a pusher is used to sort a cluster of objects that lie on a table into corresponding target regions. In this environment, we generate a dataset consisting of 150,000 transition pairs with two objects using Pymunk. Following the success of previous graph-based dynamics models [4, 37, 36], we use GNNs as the model class. The model takes the object positions as input and allows compositional generalization to extrapolate settings containing more objects, supporting up to 8 objects as tested in our benchmark.

- **Rope manipulation.** Figure 4b shows the task of manipulating a deformable rope into a target shape. We generate a dataset of 60,000 transition pairs through random interactions using Nvidia FleX [45]. We use an MLP to model the dynamics, and the observation $y_t$ is the position of four selected keypoints on the rope.

### 4.1 How does the varying number of ReLUs affect the prediction accuracy?

**Recover the ground truth piecewise affine system from data.** The sparsification procedure starts with the full model with four hidden layers and 576 ReLU units. It then undergoes seven iterations of sparsification, with the number of remaining ReLUs, represented as $\varepsilon_k$, diminishing from 25 down to 2. As illustrated in Figure 2, the sparsified model, which retains only two ReLUs, accurately identifies the region partition and achieves a nearly zero distance from the ground truth. This enables

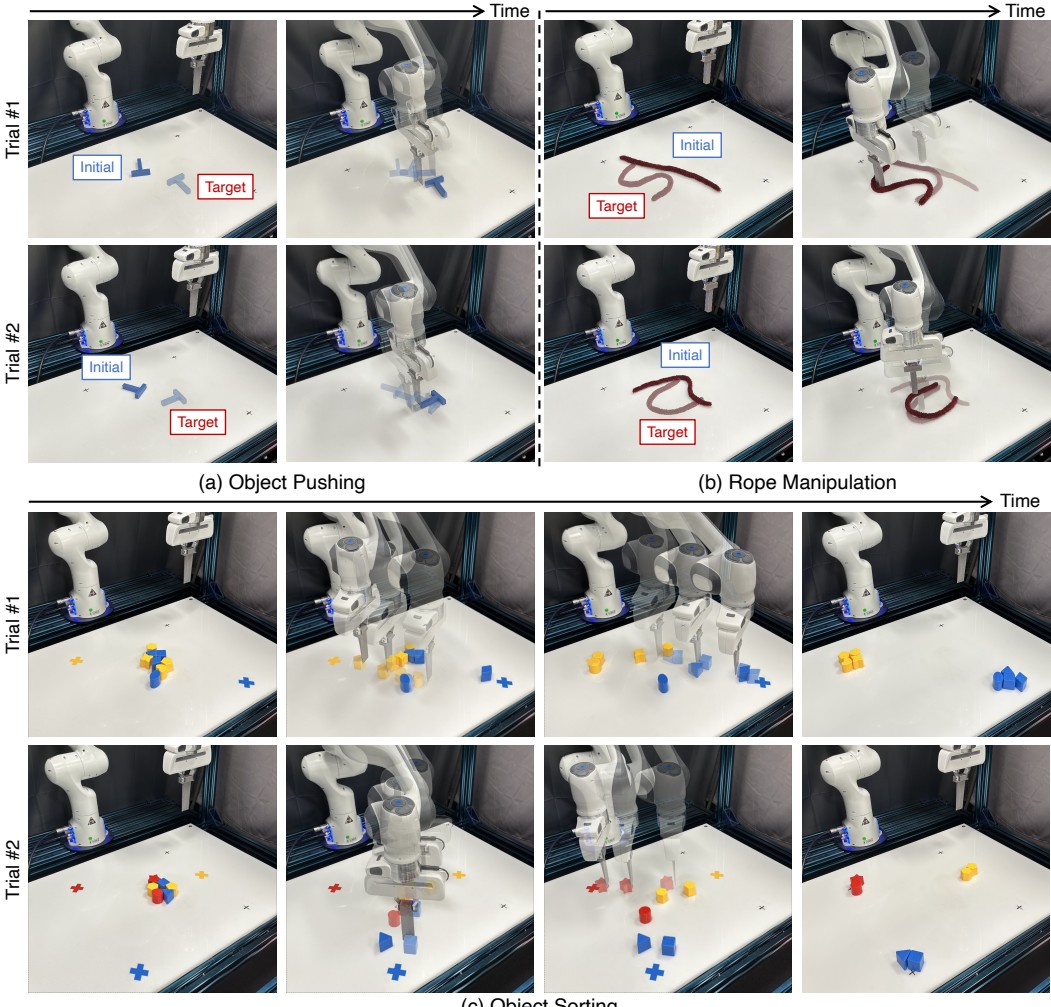

Figure 4: **Qualitative results on closed-loop feedback control.** (a) In object pushing, the objective is to manipulate the object to reach a randomly generated target pose, depicted as transparent in the first column. The second column illustrates how the planner, using the sparsified model, can leverage feedback from the environment to compensate for the modeling errors and accurately achieve the target. (b) The framework is also applicable to rope manipulation. Our sparsified model, in conjunction with the MIP formulation, facilitates closed-loop feedback control to manipulate the rope into desired configurations. (c) Our framework also consistently succeeds in object sorting tasks that involve complex contact events. Using the same model with the MIP formulation, the system can manipulate up to eight objects, sorting them into their respective regions.

the model to recover the underlying ground truth system and demonstrates the effectiveness of the sparsification procedure.

**Future prediction using sparsified models at different sparsification levels.** Existing literature provides comprehensive studies indicating that neural networks are overparameterized [22, 23, 16]. Still, we are interested in understanding how the proposed sparsification process affects the model prediction accuracy. We evaluate our framework on three benchmark environments, object pushing, sorting, and rope manipulation. Starting with the full model, we gradually sparsify it using decreasing values of $\varepsilon_k$. During training, we focus solely on the accuracy of one-step prediction but evaluate the models for their long-horizon predictive capability.

Figure 3a illustrates the prediction accuracy for models with varying numbers of ReLUs. "Object Sorting-2" denotes the task of sorting objects into two piles, while "Object Sorting-3" represents sorting into three piles. The blue curve shows the full-model performance, and the shaded area denotes the region between the 25th and 75th percentiles over 100 trials. The figure suggests that, even with significantly fewer ReLUs, the model still achieves a reasonable future prediction performance, with the confidence region significantly overlapping with that of the full model. It is worth

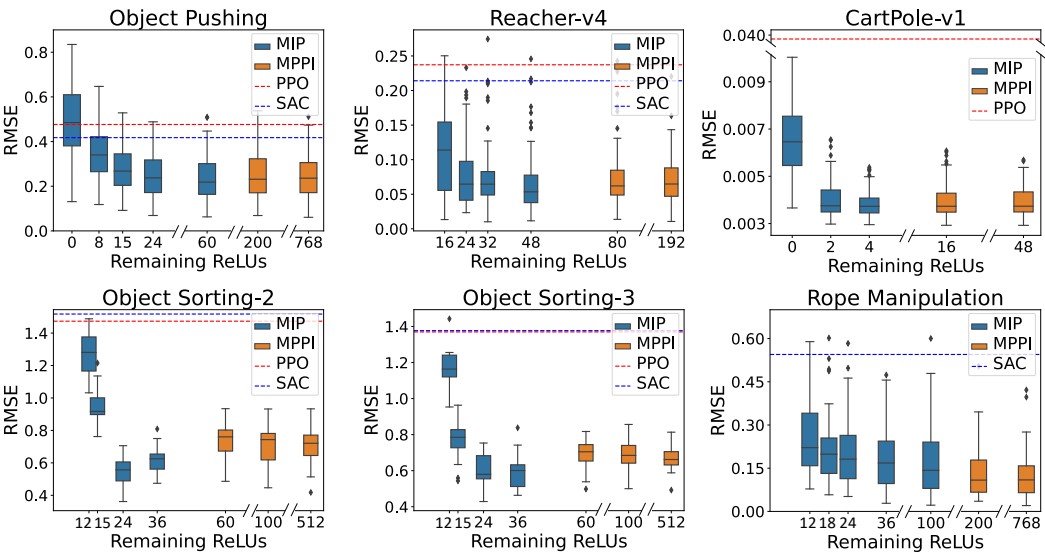

Figure 5: **Quantitative analysis of model sparsification vs. closed-loop control performance.** The horizontal axis represents the number of ReLUs remaining in the model, and the vertical axis indicates the closed-loop control performance. As shown in the figure, there exists a nice trade-off between the levels of model sparsification and the performance of closed-loop control. Models with fewer ReLUs are typically less accurate than the full model but make the MIP formulation tractable to solve. Across the spectrum of models, there exists a sweet spot, where a model, although only reasonably accurate, benefits from more powerful optimization tools and can lead to superior closed-loop control results. Moreover, our method consistently outperforms commonly used RL techniques such as PPO and SAC.[‡]

noting that our framework is adaptable to both vanilla MLPs (utilized in object pushing and rope manipulation) and GNNs (employed for object sorting), thereby showcasing the broad applicability of our proposed method. Later in Section 4.2 and 4.3, we will demonstrate that the sparsified models, although slightly less accurate than the full model, can yield superior open-loop and closed-loop optimization results when paired with more effective optimization tools.

## 4.2 How does the varying number of ReLUs affect open-loop planning?

Having obtained the sparsified models and examined their prediction accuracy, we next assess how these models can be applied to solve the trajectory optimization problem in Equation 10. The sparsified model contains significantly fewer ReLUs, making it feasible to use formulations with better optimality guarantees, as discussed in Section 3.5. Specifically, we formulate the optimization problem using MIP (Equation 11) and solve the problem using a commercial optimization solver, Gurobi [18]. We compare our method with MPPI, a commonly-used sampling-based alternative from the model-based RL community. As illustrated in Figure 3b, the MIP formulation permits the use of advanced branch-and-bound optimization procedures. With a sufficiently small number of ReLU units remaining in the neural dynamics models, we can solve the problem optimally. This consistently outperforms MPPI by a significant margin.

## 4.3 Can the sparsified model deliver better closed-loop control results?

The results in Section 4.2 only tell us how good different optimization procedures are as measured by the learned dynamics model. However, what we really care about is the performance when executing optimized plans in the original simulator or the real world. Therefore, it is crucial to evaluate the effectiveness of these models within a closed-loop control framework. Here we employ an MPC

---

[‡]We omit the result of PPO on rope manipulation due to compute limitations, because our rope simulator does not support accelerated-time simulation and takes excessively long before PPO gains reasonable performance. We omit the result of SAC on `CartPole-v1` because the Stable Baselines 3 SAC implementation does not support a discrete action space.

framework that, taking into account the feedback from the environment, allows the agent to make online adjustments to the action sequence.

Figure 4 visualizes multiple execution trials of object pushing, sorting, and rope manipulation in the real world using our method. Our framework reliably pushes the object to its target pose, deforms the rope into the desired shape, and sorts the many objects into the corresponding piles. We then present the quantitative results for object pushing, sorting, and rope manipulation, along with two tasks from OpenAI Gym [7], `CartPole-v1` and `Reacher-v4`, measured in simulation, in Figure 5. Across various tasks, we observe a similar trade-off between the levels of model sparsification and closed-loop control performance. As the number of ReLUs decreases, there is typically a slight decrease in prediction accuracy, but as illustrated in Figure 5, this allows us to formulate the trajectory optimization problem as an MIP and solve it using efficient branch-and-bound algorithms. Consequently, within the spectrum of sparsified models, there exists an optimal point where a model, albeit only reasonably accurate, benefits from the more effective optimization tools and can result in better closed-loop control performance. Our iterative sparsification process, discussed in Section 3.4, enables us to easily identify such model. Furthermore, our method consistently outperforms commonly used RL techniques such as PPO [59] and SAC [19] when using the same number of interactions with the underlying environments.

## 5   Discussion

**Conclusion.** In this work, we propose to sparsify neural dynamics models for more effective closed-loop, model-based planning and control. Our formulation allows an end-to-end optimization of both the model class and the weight parameters. The sparsified models enable the use of efficient branch-and-bound algorithms and can deliver better performance in closed-loop control. Our framework applies to various dynamical systems and multiple neural network architectures, including vanilla MLPs and complicated GNNs. We also demonstrate the effectiveness and applicability of our method through its application to simple piecewise affine systems and manipulation tasks involving complex contact dynamics and deformable objects.

Our work draws inspiration and merges techniques from both the learning and control communities, which we hope can spur future investigations in this interdisciplinary direction to take advantage and make novel use of the powerful tools from both communities.

**Limitations and future work.** Our method relies on sparsifying neural dynamics models to fewer ReLU units to make the control optimization process solvable in a reasonable time due to the worst-case exponential run time of MIP solvers. Although our experiments showed that this already enabled us to complete a wide variety of tasks, our approach may struggle when facing a much larger neural dynamics model.

Our experiments also demonstrated superior closed-loop control performance using sparsified dynamics models with only reasonably good prediction accuracy as a result of benefiting from stronger optimization tools, but our approach may suffer if the sparsified dynamics model becomes significantly worse and incapable of providing useful forward predictions.

**Acknowledgments.**   This work is in part supported by ONR MURI N00014-22-1-2740. Ziang Liu is supported by the Siebel Scholars program.

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

# A How does our method compare to prior works in model-based RL?

In this experiment, we aim to examine how the closed-loop control performance of our method compare to prior works in model-based reinforcement learning, evaluated on standard benchmark environments. We conduct experiments on two additional tasks from OpenAI Gym [7], `CartPole-v1` and `Reacher-v4`, following the same procedures as described in Section 4.3. On top of a sampling-based planner (MPPI) and a model-free RL method (PPO), we employ two additional model-based RL methods, (1) using PPO to learn a control policy from our learned full neural dynamics model, and (2) MBPO [29] learning a model and a policy from scratch. The model-based RL methods require additional time to train a policy using the learned dynamics model, whereas our approach directly optimizes a task objective over the dynamics model without needing additional training.

The experiment results shown in Figure 6 further demonstrate the superior performance of our approach compared to prior methods on the two standard benchmark tasks. Notably, our approach achieves better performance with highly sparsified neural dynamics models with fewer ReLUs compared to prior works.

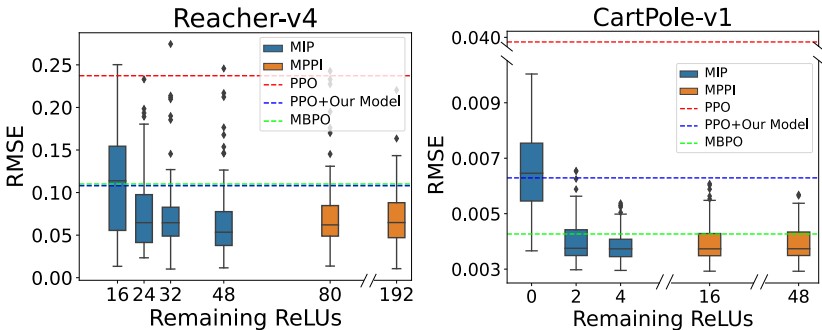

Figure 6: Closed-loop control performance of our method (MIP) compared against prior methods on two new environments. Our method with fewer ReLUs outperforms prior methods using models with more ReLUs, and we similarly observe a sweet spot that balances between model prediction accuracy and control performance.

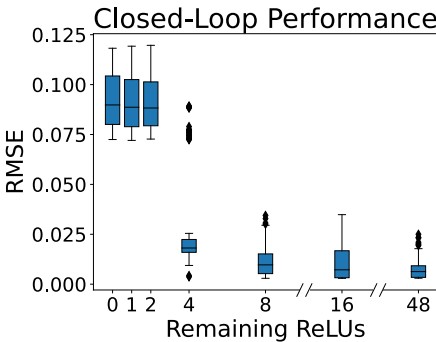

Figure 7: We tested the closed-loop control performance of dynamics models trained and simplified using our method by incorporating them as the forward model in a model-based RL framework optimized with PPO. Our findings indicate that even when the dynamics models are substantially simplified, they continue to allow for satisfactory control performance.

# B Do models trained using our approach generalize to prior model-based RL methods?

The neural dynamics model learned in our method is generic and not limited to only working with our planning framework. We take the learned full and sparsified dynamics models on the `CartPole-v1` environment and train a control policy with PPO interacting only with the learned model, and provide the experiment results in Figure 7.

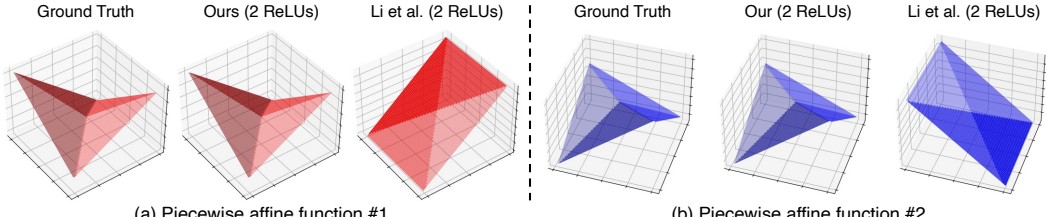

|  |  |
|---|---|
| (a) Piecewise affine function #1 | (b) Piecewise affine function #2 |

Figure 8: **Comparison with the channel pruning baseline on recovering the PWA functions.** We evaluate both our sparsification pipeline and the channel pruning baseline using two hand-designed piecewise affine functions, each composed of four linear pieces. Our pipeline successfully recovers the underlying ground truth system, whereas the baseline does not provide an accurate match.

The results show that the neural dynamics models trained in our method can generalize and combine with another model-based control framework. As the model becomes progressively sparsified, the closed-loop control performance gracefully degrades.

## C  How does our sparsification technique compare to prior neural network pruning methods?

The aim of this study is to discuss and compare our sparsification technique with the pruning methods commonly employed in the field. Most pruning strategies in existing literature primarily focus on eliminating as many *neurons* as possible to reduce computation and enhance efficiency. In contrast, our method aims to eliminate as many *nonlinearities* from the neural networks as possible. This differs from channel pruning, which only zeroes out values. Our approach permits the replacement of ReLU activations with identity mappings, the inclusion of which allows a more accurate model to be achieved at an equivalent level of sparsification. This offers a considerable advantage during the planning stage.

To illustrate our point more concretely, we provide, in this section, experimental results comparing our method against Level 1 Channel Pruning as referenced in [35].

### C.1  Evaluation on Piecewise Affine (PWA) Functions

The two pruning methods are tasked to recover the ground truth PWA functions, as detailed in the experiment section of the main paper. Figure 8 illustrates the results after sparsifying the neural networks to two rectified linear units (ReLU) using both methods. Our method successfully identifies the region partition and the correct equations for describing values of each region, whereas the baseline [35] exhibits noticeable deviations.

### C.2  Evaluation on Dynamics Prediction

In this section, we extend the comparison to three other tasks: object pushing, object sorting, and rope manipulation. For the object pushing and rope manipulation tasks, we train the neural dynamics model for a defined number of epochs before pruning is carried out by masking particular channels. Post-pruning, model speedup is performed using Neural Network Intelligence Library [51] to alter the model's architecture and remove masked neurons. This process is repeated as further channels are pruned and the models are finetuned for additional epochs.

For the object sorting task involving graph neural networks, we perform a similar procedure to construct the baseline. During the initial model training phase, the mask resulting from the $L_1$ norm pruning operation is used to nullify specific weights, and the corresponding gradients are also masked during the finetuning phase.

To ensure fairness and reliability in the comparison, we maintain identical settings for both our sparsification technique and the pruning baseline. Therefore, for every round of compression, both models are subjected to the same number of training epochs, using the same dataset, and are reduced to the same number of ReLU units.

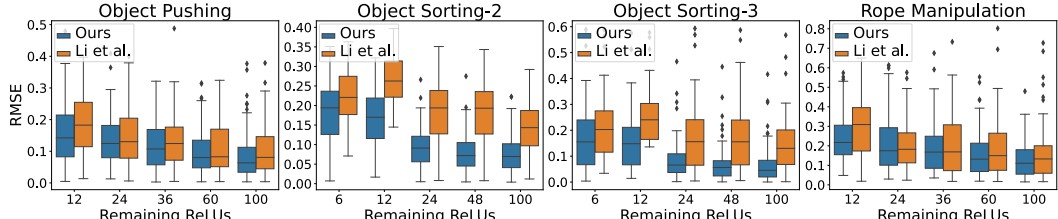

Figure 9: **Dynamics prediction error of sparsified models using our method vs. baseline.** We compare the dynamics prediction error of models sparsified using our method against models sparsified using the channel pruning method proposed by Li et al. [35]. The x-axis represents the number of remaining ReLU units in the model. The y-axis represents the prediction error measured by the root mean squared error between the prediction and the ground truth next state. Because our sparsification method only targets non-linear units while allowing linear units, models sparsified using our method constantly exhibit lower prediction error across all task settings.

We provide quantitative comparisons between our sparsification method and the baseline in Figure 9. Throughout the sparsification process, because our sparsification objective allows replacing non-linearities with identity mappings, our method consistently achieves a superior performance measured by prediction error, across all tasks.

## C.3 Evaluation on Closed-Loop Control

In this experiment, we aim to further examine whether our method also boosts the closed-loop control performance that is critical for executing the optimized plans in the real world. We choose the object pushing task, and prune the learned dynamics model down to 36, 24, 18, and 15 ReLU units using our proposed sparsification method and the channel pruning method proposed by Li et al. [35] respectively. As shown in Figure 10, models pruned using our method consistently exhibit superior closed-loop control performance across all sparsification levels.

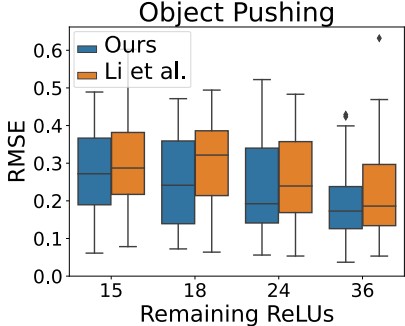

Figure 10: An ablation of our sparsification method compared with a prior network pruning method, evaluated by closed-loop control performance, demonstrating superior performance in closed-loop control.

