# OpenReview forum: "Model-Based Control with Sparse Neural Dynamics"
_NeurIPS.cc/2023/Conference — NeurIPS 2023 poster_

### Official Review · Reviewer_LeuW · 2023-07-03

**Soundness:** 2 fair
**Presentation:** 3 good
**Contribution:** 2 fair
**Rating:** 5
**Confidence:** 3

**Summary:**

This paper proposes a method for pruning neural networks with ReLu activation functions during training. When employed for learning the dynamics of a control system, this often leads to networks with few activation functions performing similarly as large networks. This allows to apply mixed integer programming techniques to determine optimal control policies. The effectiveness of this approach is demonstrated in a comparison with a sampling based optimizer on several tasks for a robotic manipulator.

**Strengths:**

The idea of formulating planning problems with neural network dynamics as mixed integer program seems novel to me. Moreover, I find the proposed approach of reducing the number of neurons during training very interesting, but I cannot comment on its novelty since I am no expert on this topic. Overall, the paper reads very well and has a good structure. The nice demonstration of the proposed method in a real-world robotic experiment is also a strength of the paper.

**Weaknesses:**

My biggest concern is the missing discussion of computational complexity/computation time. I am not an expert on mixed integer programming, but a quick search suggests that even mixed integer linear programming is already NP hard, so this seems to be a problem in general. I understand that this problem can be mitigated through a sufficiently small number of neurons (‘If only a few ReLUs are left in the model, Equation 10 can be efficiently solved to global optimality’), but I am missing a clear specification what sufficient means in this context. Moreover, I would expect it to be connected to the prediction horizon, i.e., the number of time steps considered in the sum in (10). Therefore, I think a complexity (e.g., in O notation) should be provided to give the reader an impression how severe the computation times grows. This weakness also limits the usefulness of the evaluation in my opinion. Why would you reduce the number of neurons for MPPI in practice? I think the performance in relation to computation time is rather the important metric to look at for this comparison, i.e., a complexity-performance trade-off comparison would be essential. Therefore, the comparison seems a little unfair at the moment. This similarly extends to the robot experiment in Sec. 4.3, where the sampling and the horizon rate of the MPC are not specified. Moreover, it is not clear how fast the robot moves. These are all parameters which crucially influence how challenging the problem is.

In general, the method seems tailored to piece-wise affine systems or systems that look almost like that. The examples seem to go into this direction, but I cannot say for sure since I could not find information what dynamics are actually learned in the robotic examples. I think it would be interesting to see how many activation functions are needed for accurately learning highly nonlinear dynamics, e.g., cart-pole swing-up, and how this affects the control performance.

Overall, the novelty seems to be mainly the (straightforward) connection of existing ideas, even though I admit that it is a very clever combination. The proposed reduction of neurons is not even targeted to achieve a high control performance, but only to maintain high model accuracy. High control performance is only addressed a posteori by selecting the best model observed in experiments. Doing it as described in lines 206-208 runs the risk of executing a potentially bad controller on a real system. This seems like a dangerous thing to generally do. I think it would be much more interesting to directly optimize the model for achieving the best control performance.

Finally, I find the comparison in the numerical evaluation a little weak. When no experiment has to be done like for the open-loop planning performance evaluation, I think more methods than only MPPI should be investigated. I am not an expert, but I know there exists more than one planning/control method for nonlinear systems, e.g., MPC with NN dynamics (Salzmann et al., 2023). Moreover, I do not understand why the learned dynamics model is not used as environment for the classical RL methods. It is apparently accurate enough to allow for a direct transfer to a real-world experiment. So why are the interactions limited to a relatively small number with the real environment, when a large number of interactions with a potentially only slightly more inaccurate model are available?

T. Salzmann, E. Kaufmann, J. Arrizabalaga, M. Pavone, D. Scaramuzza and M. Ryll, "Real-Time Neural MPC: Deep Learning Model Predictive Control for Quadrotors and Agile Robotic Platforms," in IEEE Robotics and Automation Letters, vol. 8, no. 4, pp. 2397-2404, April 2023

**Questions:**

What is the point of the regularization in (9) if you bound the total number of ReLu activation functions through $\epsilon$ in (5) anyways?

What is a closed-loop planning performance? Or do you mean closed-loop control performance?

In (8), is $z_{ij}$ the approximation of $\pi_{ij}$?

**Limitations:**

Potential negative societal impact is not mentioned in the paper. Some limitations of the proposed approach (e.g., restriction to ReLu activation functions) are mentioned throughout the paper, but there is no dedicated paragraph about limitations. I think there are some limitations (e.g., computational complexity) that should be added.

---

> ### Author Rebuttal · Authors · 2023-08-10
>
> Thank you for your time reviewing our paper, and for your constructive suggestions that helped improve our work.
>
> > I think a complexity (e.g., in O notation) should be provided to give the reader an impression how severe the computation times grows.
>
> We agree with the reviewer that a more precise description of the runtime is useful for the readers. The complexity is exponential in the number of ReLUs in the worst case, but in practice solvers like Gurobi are highly optimized, and their branch-and-bound implementations can solve medium-scale problems very quickly, although the performance highly depends on the problem formulation, problem conditioning, and the tightness of the convex relaxation.
>
> > I think the performance in relation to computation time is rather the important metric to look at for this comparison, i.e., a complexity-performance trade-off comparison would be essential.
>
> We agree that a complexity-performance trade-off is essential, and we used the number of remaining ReLUs in the dynamics model as a proxy for computation complexity. The solve time in practice depends on the problem formulation, conditioning, initialization, and various solver techniques and heuristics. In our experiments, solving a MIP with Gurobi induces a high variance in computation time across different tasks, dynamics model architectures, and problem initializations, as shown in Rebuttal PDF Fig. 3 left.
>
> Although the wall clock computation time exhibits high variance for the same neural dynamics model, we observe that the computation time is positively correlated with the number of ReLUs in the network, and thus chose to use the number of remaining ReLUs as an effective proxy for computation complexity in our experiments.
>
> > the method seems tailored to piece-wise affine systems or systems that look almost like that
>
> Our method focuses on neural networks with ReLU activations, which act as piecewise affine functions. With a sufficient number of pieces, they can exhibit extremely strong approximation power and can be used to approximate nonlinear functions that are smooth arbitrarily well. Piecewise affine systems have been widely studied and used in the literature [Sontag et al.] to approximate highly complex and nonlinear systems.
>
> Our experiments examined the applicability of our approach to systems that are not piecewise affine. The dynamics of the rope in our Rope Manipulation task are not piecewise affine. The Object Pushing task involves modeling the object’s orientation, which is also not piecewise affine.
>
> > The proposed reduction of neurons is not even targeted to achieve a high control performance, but only to maintain high model accuracy.
>
> Using the same control optimization tool, we believe that a dynamics model with higher accuracy leads to better control performance as long as the model size does not need to be too large for the control optimization tool to solve within reasonable time. Our experiments shown in Fig 5 and Rebuttal PDF Fig. 1 also show that the closed-loop control performance decreases with the model accuracy, using the same optimization tool.
>
> We would also like to note that targeting to simultaneously achieve a high control performance during sparsification might necessitate a differentiable design for the control optimization procedure, which we will leave for future work.
>
> > I think more methods than only MPPI should be investigated
>
> We provided comparisons to MPPI, as well as model-free RL methods (PPO and SAC), all of which are widely used in the literature solving similar control and manipulation tasks. We also performed additional experiments using model-based RL (MBPO [Janner et al.]) on two OpenAI Gym benchmark environments (Cartpole-v1, Reacher-v4). The results are shown in Rebuttal PDF Fig. 1.
> Our approach outperforms prior methods on the two standard RL benchmark environments. Notably, our approach achieved superior performance with highly sparsified neural dynamics models with fewer ReLUs compared to prior works.
>
> > I do not understand why the learned dynamics model is not used as environment for the classical RL methods.
>
> The neural dynamics model learned in our method is generic and not limited to only working with our planning framework. We took the learned full and sparsified dynamics models and trained a control policy with PPO interacting only with the learned dynamics model as suggested, and provide the experiment results below (also included in Rebuttal PDF Fig. 2).
>
> | Num. ReLUs |    48    |    16    |     8    |     4    |     2    |     1    |     0    |
> |:----------:|:--------:|:--------:|:--------:|:--------:|:--------:|:--------:|:--------:|
> |    RMSE    | 0.006291 | 0.007201 | 0.009670 | 0.018122 | 0.088279 | 0.088655 | 0.089807 |
>
> The results above on Cartpole-v1 show that the neural dynamics models trained in our method can generalize and combine with another model-based control framework. As the model becomes progressively sparsified, the closed-loop control performance gracefully degrades. Our best performing model, MIP with 4 ReLUs, outperformed these models and achieved an RMSE of 0.003728.
>
> > why are the interactions limited to a relatively small number with the real environment, when a large number of interactions with a potentially only slightly more inaccurate model are available?
>
> We limited the number of interactions for training with the two model-free methods, PPO and SAC, to be the same as the number of real environment interactions that our method used to train the dynamics model. This setting is practical in scenarios where interacting with the real environment is costly or not always available, and is widely studied in the RL community. As the reviewer suggested, we conducted additional experiments using a model-based RL method to interact with the neural dynamics model to learn a control policy, reported above.
>
> **Due to length limit, we are happy to respond to the remaining questions during the discussion phase.**

---

> ### Author Response · Authors · 2023-08-10
> **Additional Author Responses to Reviewer LeuW**
>
> > Why would you reduce the number of neurons for MPPI in practice?
>
> In the closed-loop control results in Fig. 5, we aimed to demonstrate that our method achieves comparable or better performance with a highly sparsified model, compared to MPPI using a full, non-sparsified model (rightmost column of each plot). We included the results of MPPI with slightly sparsified models to illustrate how a slightly sparsified model affects closed-loop control performance using the same optimizer, and to visualize the trend that a sweet spot exists where a reasonably accurate model can benefit from more powerful optimization tools, leading to superior closed-loop control performance.
>
> > This similarly extends to the robot experiment in Sec. 4.3, where the sampling and the horizon rate of the MPC are not specified.
>
> In the Object Pushing, Object Sorting, and Rope Manipulation tasks, we empirically find that using a horizon of one and re-optimizing at each step based on environmental feedback was sufficient to successfully complete a variety of tasks.
>
> > it is not clear how fast the robot moves
>
> We provided videos of our method controlling a real robot performing the tasks in the supplementary video. Since each action is standalone, there is no additional feedback from the environment while an action is being executed, so the speed of the robot does not affect the task completion.
>
> > it would be interesting to see how many activation functions are needed for accurately learning highly nonlinear dynamics, e.g., cart-pole swing-up, and how this affects the control performance
>
> We showed in Fig. 3a how the number of remaining ReLUs affects open-loop prediction accuracy of the learned dynamics model. For example, in Object Pushing, a model with 0 ReLUs (linear) failed drastically, while a model with 8 ReLUs was still reasonably accurate.
>
> In the Reacher-v4 environment, we observe a gap where all models with 16 ReLUs or less fail significantly. In the Cartpole-v1 environment with simpler dynamics, neural dynamics models sparsified down to 1 or 2 ReLUs can still have reasonable performance, but when sparsifying down to 0 ReLUs (linear), we observe a significant drop in both prediction accuracy and closed-loop control performance.
>
> Below we show the results on the closed-loop control performance of models sparsified to different numbers of ReLUs remaining on the Reacher-v4 environment.
>
> | Num. ReLUs |    48   |    32   |    24    |    16    |     8    |     2    |
> |:----------:|:-------:|:-------:|:--------:|:--------:|:--------:|:--------:|
> |    RMSE    | 0.053467 | 0.064591 | 0.064659 | 0.113871 | 0.153998 | 0.170154 |
>
> While the control performance gracefully degrades with the number of ReLUs in the beginning when the number of ReLUs is still sufficient to learn a reasonably accurate model, we start to observe significantly worse control performance once the number of ReLUs becomes insufficient to represent the complex dynamics of the system.
>
> > What is the point of the regularization in (9) if you bound the total number of ReLu activation functions through $\epsilon$ in (5) anyways?
>
> We indeed use $\epsilon$ as a cap on the number of ReLUs. The regularization term in (9) aims to discourage unnecessary use of ReLU and identity mappings so that the gradient-based optimization favors a model architecture that is more likely to perform well after we sparsify the model to fewer ReLUs in the next round. For example, we start with a full model of 512 ReLUs in the first iteration, and aim to sparsify down to 64 ReLUs after the first iteration. Because of the redundancy in neural networks, the model can likely achieve similar performance with all 512 ReLU activations, or with fewer ReLUs. If the model chooses to have 512 or significantly more than 64 ReLUs in the first iteration, then after we prune the network down to 64 ReLUs for the next iteration, this drastic change in modeling capacity could lead to noticeably larger decrease in prediction accuracy.
>
> >What is a closed-loop planning performance? Or do you mean closed-loop control performance?
>
> Yes, we apologize for the confusion.
>
> >In (8), is $z_{ij}$ the approximation of $\pi_{ij}$?
>
> Yes, $z_{ij}$ is the continuous softmax approximation of the discrete probability distribution $\pi_{ij}$.
>
> > I think there are some limitations (e.g., computational complexity) that should be added.
>
> Please refer to the global rebuttal response for discussions of limitations of our work.
>
> **Have these responses addressed the reviewer’s concerns? We look forward to continuing the discussion.**

---

> > ### Comment · Reviewer_LeuW · 2023-08-14
> >
> > Thank you for this detailed response. I appreciate the additional simulations and think they would significantly improve the paper. In particular, the usage of MIP indeed seems to provide a slight advantage to every other method. Despite these changes, one of my major concerns still remains: It is not clear how other methods perform when comparing them for the same computation time. This cannot be resolved by using the number of remaining neurons as an approximate measure for this since the methods' computation time depends differently on the number of neurons. Therefore, a plot showing computation time over performance is critical to demonstrate the advantages of the proposed method for closed-loop control, where the computation time defines the achievable sampling rate. Therefore, I currently do not intend to change my scores.
> >
> > Follow-up question:
> >
> > "Since each action is standalone, there is no additional feedback from the environment while an action is being executed"
> > $\rightarrow$ Does this mean you only use the model and MIP optimization to plan a trajectory? If so, this should be stated more clearly.

---

> > > ### Author Response · Authors · 2023-08-18
> > > **Additional Experiment Results Requested by Reviewer LeuW**
> > >
> > > Thank you for the follow up questions! We provide the experiment results requested and our response below.
> > >
> > > > a plot showing computation time over performance is critical to demonstrate the advantages of the proposed method for closed-loop control
> > >
> > > This is a great point. Following your suggestion, **we conducted additional experiments examining the tradeoff between** ***computation time*** **and** ***closed-loop control performance*** using the Reacher-v4 environment. We are unfortunately unable to submit plots in the discussion stage, so we present the results in the tables below.
> > >
> > > The first row of each table denotes the mean computation time in seconds, and the second row denotes the closed-loop control performance measured by RMSE. The first table contains results using our method (MIP), and the second table contains results using MPPI.
> > >
> > > Both methods achieved better closed-loop control performance when given more time to compute. **Our method (MIP) improved significantly in closed-loop control performance with more computation time**, leading to an superior RMSE of 0.04795 with 0.09716 seconds of computation. On the contrary, **the performance of MPPI plateaued with more computation time** and achieved an RMSE of 0.06572 with 0.100086 seconds of computation.
> > >
> > > **MIP (Ours), 48 ReLUs**
> > > | | | | | | |
> > > |--|--|--|--|--|--|
> > > | Time (s) | 0.01198 | 0.02257 | 0.04903 | 0.08007 | **0.09716** |
> > > | RMSE | 0.20975 | 0.19609 | 0.17762 | 0.06916 | **0.04795** |
> > >
> > >
> > > **MPPI, 192 ReLUs**
> > > | | | | | | |
> > > |--|--|--|--|--|--|
> > > | Time (s) | 0.00791 | 0.01822 | 0.04686 | 0.08284 | 0.100086 |
> > > | RMSE | 0.09273 | 0.07185 | 0.06868 | 0.06693 | 0.06572 |
> > >
> > > **This comparison further supports that our method, using a sparsified dynamics model, can benefit from more powerful optimization tools, and lead to superior closed-loop control performance under the same computation time compared to MPPI using a full dynamics model.**
> > >
> > >
> > >
> > > > "Since each action is standalone…"  Does this mean you only use the model and MIP optimization to plan a trajectory? If so, this should be stated more clearly.
> > >
> > > **Our method performs closed-loop control by incorporating environment feedback between actions, but not during an action execution.** For example, in the Object Sorting tasks, after a push action has been completed, we use the new environment state observed to optimize for the next action, but we do not account for environment feedback while the robot is executing a single push action from start to end. We apologize for the confusion.
> > >
> > > We hope that our additional experiments and responses addressed the reviewer’s questions and concerns. **We would be happy to continue the discussion if the reviewer has additional questions or concerns.**

---

> > > > ### Comment · Reviewer_LeuW · 2023-08-20
> > > >
> > > > Thank you for these clarifications and the additional simulation results. I believe these results will definitely improve the paper and recommend including them in the main paper. As these results show that the method offers clear advantages in scenarios, where a comparatively large time for the computation of control inputs is available, I suggest indicating this in the paper.
> > > >
> > > > Due to these additional results, I have updated my score.

---

### Official Review · Reviewer_fLUK · 2023-07-06

**Soundness:** 2 fair
**Presentation:** 4 excellent
**Contribution:** 4 excellent
**Rating:** 5
**Confidence:** 3

**Summary:**

This paper proposes a new framework for model-based control. The approach focuses on learning a sparse deep neural network and using a mixed-integer program solver for closed-loop planning. Experimental results are presented on several tasks including object and rope manipulation tasks. The results show that the proposed approach improves performance over strong baselines.

**Strengths:**

- The framework proposed in this paper elegantly combines concepts from deep neural network (DNN) pruning and mixed integer programming (MIP) into a solution for model-based control.
- The specific technique for sparsifying a DNN by removing non-linearities rather than simply dropping nodes intuitively matches the goals of eventually using the model with a MIP solver.
- The experiments demonstrate that the approach is effective in both simulation and in the real-world on real hardware -- providing strong evidence that this is a generally applicable approach.
- The technique is demonstrated with both MLP and GNN based models, showing the versatility of the proposed approach.
- The supplemental video presents strong qualitative evidence supporting the efficacy of the approach on real-robots.
- The paper is very well written.

**Weaknesses:**

- It is unclear if the performance gains in Figure 5 are significant. On the Object Pushing and Rope Manipulation tasks MIP does not appear to outperform MPPI. And the improvements of MIP on the Object Sorting tasks appear to be within the error bars.
- A key ablation is missing. One of the main claims in this work is that the proposed sparsification technique, which focuses on removing nonlinearities rather than neurons, improves performance. Only a partial ablation in support of this claim is provided in the appendix A2. Specifically, A2 shows that the proposed approach leads to lower prediction error. However, to quote L277-279 in the main paper, “what we really care about is the performance when executing optimized plans in the original simulator or the real world. Therefore, it is crucial to evaluate the effectiveness of these models within a closed-loop control framework.” I agree. Thus, it is similarly crucial to perform this ablation in a closed-loop setting.

**Questions:**

I am willing to adjust my rating if these questions can be answered:

- Please comment further on the results in Figure 5; there is a trend but why is this a significant advance?
- Does the proposed sparsification technique improve closed-loop performance?
- What are the limitations of the work?

**Limitations:**

No, the paper does not discuss limitations.

---

> ### Author Rebuttal · Authors · 2023-08-10
>
> Thank you for reviewing our paper, and for your insightful feedback that helped improve our work.
>
> > Please comment further on the results in Figure 5; there is a trend but why is this a significant advance?
>
> In both Object Sorting tasks, our method using MIP and a sparsified model of 24 and 36 ReLUs demonstrate superior performance compared to the method using MPPI and a full model with 512 ReLUs, indicated by the lower interquartile range and median, supporting our statement that having a sparsified model allows us to benefit from more powerful optimization tools, leading to superior closed-loop control results despite having a less accurate dynamics model.
>
> In Object Pushing, MIP with a model of 60 ReLUs achieved comparable performance to MPPI with a model of 768 ReLUs, and slightly better performance compared to MPPI with 200 ReLUs.
>
> We agree that MIP achieves a similar performance to MPPI on Rope Manipulation. There is a more significant gap in model prediction error between the full dynamics model and highly sparsified models due to the complex physical properties of the deformable rope (Fig. 3a). The task requires accurately manipulating the rope to match a target shape specified with key points along the rope, thus the performance is highly dependent on having an accurate dynamics model. Even though using a sparsified model enabled us to leverage a more powerful optimization tool giving better results in open-loop optimization (Fig. 3b), the model prediction error limits the performance upper bound in closed-loop planning. Future improvements could involve the co-optimization of the model sparsification procedure with control synthesis. Enabling the sparsification process to be mindful of the subsequent control task may lead to even better performance
>
> > Does the proposed sparsification technique improve closed-loop performance?
>
> The reviewer pointed out that it is crucial to perform an additional ablation study of our proposed sparsification technique in a closed-loop setting, as we also stated in the paper. To that end, we performed additional experiments to validate the performance again on closed-loop control of our proposed method, and provide the results below (also in Rebuttal PDF Fig. 3 right).
>
> | Num. ReLUs |    36    |    24    |    18    |    15    |
> |:----------:|:--------:|:--------:|:--------:|:--------:|
> |    Ours    | 0.172873 | 0.192272 | 0.241469 | 0.271934 |
> |  Li et al. [1] | 0.186144 | 0.239412 | 0.321603 | 0.287213 |
>
> The numbers in each column correspond to the closed-loop control performance, measured by RMSE, using the sparsified model with the corresponding number of ReLUs.
>
> These ablation results, combined with results reported in supplemental material Fig. 2, demonstrate that our proposed sparsification scheme brings significant improvements in both open-loop prediction and closed-loop control performance.
>
> > What are the limitations of the work?
>
> Please refer to the global rebuttal response for discussions of limitations of our work.
>
> **Have these responses addressed the reviewer’s concerns? We look forward to continuing the discussion.**
>
> [1] H. Li, A. Kadav, I. Durdanovic, H. Samet, and H. P. Graf, ‘Pruning Filters for Efficient ConvNets’, CoRR, vol. abs/1608.08710, 2016.

---

> ### Author Response · Authors · 2023-08-18
> **Are there any additional questions or concerns?**
>
> We hope that our additional experiments following the reviewer’s suggestions, discussion of limitations, and responses addressed the questions and concerns raised. **We would be happy to continue the discussion if the reviewer has additional questions or concerns.**

---

> > ### Comment · Reviewer_fLUK · 2023-08-19
> > **Questions resolved**
> >
> > Thank you for the detailed response. The closed-loop performance is encouraging and I believe it will strengthen the paper. Additionally, I appreciate the discussion of limitations in the main response. I do not have any remaining questions and I have updated my score.

---

### Official Review · Reviewer_H7kE · 2023-07-06

**Soundness:** 3 good
**Presentation:** 3 good
**Contribution:** 3 good
**Rating:** 6
**Confidence:** 4

**Summary:**

This paper proposes a framework for model-based planning with forward dynamics represented as sparse neural networks. The paper examines different ways of inducing sparsity in MLP and GNN based forward models, and performs real robot manipulation experiments investigating the tradeoffs with sparsity and performance.

**Strengths:**

- the paper targets an interesting problem of learning efficient world models for control, by reducing optimization costs in learning the forward model. The specific approach of investigating how sparsity can enable this efficient learning of the forward model for real robot manipulation scenarios, is novel in my understanding.

- the paper is easy to follow, with detailed descriptions of the different ways of inducing sparsity, and the different architectures. The research questions investigated are well formulated, and adequately addressed in the experiments

- the experiments are on interesting real robot manipulation tasks like object sorting, and rope manipulation, so the findings are amenable to be deployed in real world control scenarios.

**Weaknesses:**

- the main weakness of the paper is that there aren't comparisons to prior model-based RL approaches, e.g. PETS, Dreamer, MBPO etc (https://arxiv.org/abs/1805.12114,https://arxiv.org/abs/1912.01603, https://arxiv.org/abs/1906.08253)  . If it is difficult to compare them on hardware experiments,  there should be comparisons done on simulated robot control tasks.

- it is unclear how general are the findings in terms of being applied to different model-based control frameworks. There should be a disucssion (and possibly experiments) about if we can take any prior model-based RL algorithm and replace the forward model with a sparse neural network, and see benefits under certain assumptions - if not, then a discussion of what exactly are the requirements on how the model should be learning and how a policy (or plan) should be learned is needed.

- the experiments results in my understanding are on low-dimensional states instead of images. Is there a reason for not evaluating with image observations? Encoding image observations into latent states, and then doing policy learning is a common practice in the community now - so it will be helpful to have a discussion regarding this.

**Questions:**

Please refer to my list of weaknesses above. In summary

- comparisons to prior MBRL approaches?
- discussion about generality of the findings?
- feasibility of experiments with image observations?

**Limitations:**

The limitations are a bit unclear at the moment (also refer to my list of weaknesses). It will be helpful to provide a detailed discussion of the limitations and generality of the findings.

---

> ### Author Rebuttal · Authors · 2023-08-10
>
> Thank you for reviewing our paper, and for your thoughtful comments and suggestions that have contributed to the refinement of our work.
> > there aren't comparisons to prior model-based RL approaches, e.g. PETS, Dreamer, MBPO etc
>
> We conducted additional experiments employing two model-based RL methods, 1) our learned full neural dynamics model with PPO to learn a control policy, and 2) MBPO [4] learning a model and a policy from scratch, on two OpenAI Gym [3] benchmark environments, Cartpole-v1 and Reacher-v4. Below are the results (also included in Rebuttal PDF Fig. 1)
>
> |             |           Ours          |       MPPI      | Model-free PPO | PPO + Our model |     MBPO [4]    |
> |-------------|:-----------------------:|:---------------:|:--------------:|:---------------:|:---------------:|
> | Cartpole-v1 |  **0.003728** (4 ReLUs) | 0.003735 (full) |    0.039222    | 0.006291 (full) | 0.004268 (full) |
> |  Reacher-v4 | **0.053467** (48 ReLUs) | 0.064792 (full) |    0.237225    | 0.108016 (full) |  0.110031(full) |
>
> Each number represents the median of RMSE over 100 trials. The parenthesis after the number denotes the number of ReLUs used in the neural dynamics model.
>
> The model-based RL methods require additional time to train a policy using the learned dynamics model, whereas our approach directly optimizes a task objective over the dynamics model without needing additional training.
>
> Overall, our experiment results demonstrate that our method using sparsified neural dynamics models with fewer ReLUs can be applied to a wide variety of tasks and exhibit better performance compared to prior methods examined.
>
> > it is unclear how general are the findings in terms of being applied to different model-based control frameworks
>
> The neural dynamics model learned in our method is generic and not limited to only working with our planning framework. We took the learned full and sparsified dynamics models and trained a control policy with PPO interacting only with the learned model as suggested, and provide the experiment results below (also included in Rebuttal PDF Fig. 2).
>
> | Num. ReLUs |    48    |    16    |     8    |     4    |     2    |     1    |     0    |
> |:----------:|:--------:|:--------:|:--------:|:--------:|:--------:|:--------:|:--------:|
> |    RMSE    | 0.006291 | 0.007201 | 0.009670 | 0.018122 | 0.088279 | 0.088655 | 0.089807 |
>
> The results above on Cartpole-v1 show that the neural dynamics models trained in our method can generalize and combine with another model-based control framework. As the model becomes progressively sparsified, the closed-loop control performance gracefully degrades.
>
> > Is there a reason for not evaluating with image observations?
>
> Our method is agnostic to model architectures and can be applied to incorporate image observations encoded as latent states as the reviewer suggested, as long as the encoder is a ReLU network. For the specific real-world robotic manipulation tasks we considered in this paper, prior works [1, 2] showed that using low-dimensional structured representations provides stronger generalization capabilities compared to pixel-based dynamics, especially for compositional systems containing varying numbers of objects (as is the case in our Object Sorting tasks). That’s why we chose to not evaluate with image observations in our experiments.
>
> > It will be helpful to provide a detailed discussion of the limitations and generality of the findings.
>
> Please refer to the global rebuttal response for discussions of limitations and generality of our work.
>
> **Have these responses addressed the reviewer’s concerns? We look forward to continuing the discussion.**
>
> [1] N. Watters, A. Tacchetti, T. Weber, R. Pascanu, P. W. Battaglia, and D. Zoran, ‘Visual Interaction Networks’, CoRR, vol. abs/1706.01433, 2017.
>
>
> [2] D. Driess, Z. Huang, Y. Li, R. Tedrake, and M. Toussaint, ‘Learning Multi-Object Dynamics with Compositional Neural Radiance Fields’, in Conference on Robot Learning, 2022.
>
>
> [3] G. Brockman et al., ‘OpenAI Gym’, CoRR, vol. abs/1606.01540, 2016.
>
>
> ​​[4] M. Janner, J. Fu, M. Zhang, and S. Levine, ‘When to Trust Your Model: Model-Based Policy Optimization’, CoRR, vol. abs/1906.08253, 2019.

---

> > ### Comment · Reviewer_H7kE · 2023-08-17
> > **response to rebuttal**
> >
> > Dear authors,
> >
> > Thanks for the response and the additional experiments. The comparison to model-based baselines provided is helpful. I am still not convinced by the applicability to image based observations, which is important in order to be able to tackle more realistic tasks with less assumptions on the observed states. However, I am still leaning towards accept for the paper, as my other two concerns have been partially resolved.

---

> > > ### Author Response · Authors · 2023-08-18
> > >
> > > Dear Reviewer H7kE,
> > >
> > > Thank you again for dedicating your time and effort in providing a thorough review of our paper. We deeply appreciate your constructive feedback and thoughtful comments that have helped us improve our work.

---

### Official Review · Reviewer_h4y2 · 2023-07-08

**Soundness:** 3 good
**Presentation:** 2 fair
**Contribution:** 2 fair
**Rating:** 5
**Confidence:** 3

**Summary:**

This paper focused on the combination of predictive control and model learning. An autogressive dynamic model based on a ReLU neural network is first learned over the observation space. The authors then aimed to sparsify it after introducing the indicator mapping function. To make the optimization feasible, the Gumbel-Softmax trick is applied to replace the greedy operation. Once the sparse dynamic is learned, mixed-integer programming solvers are used to obtain the control policy. Finally, the authors validated the proposed method across a few tasks and showed its promising performance.

**Strengths:**

1. The proposed framework of sparse neural dynamics in predictive control looks interesting. The use of Gumbel-Softmax to reparameterizing the original discrte optimization makes the gradient based methods feasible, and is indeed a reasonable idea.

2. The optimization for the control part is more efficient, compared with the gradient based methods, which constitutes another contribution of this paper.


**Weaknesses:**

Regarding the experimental comparison, it would be more convincing if the authors can test on commonly used reinforcement learning benchmarks. The only comparison regarding control policy is from Figure 3 (b) and it only involves different optimization solvers. The current results would leave the impression that the proposed method may overfit on these tasks only.

**Questions:**

1. What's the complexity of applying MIP?

2. What's the underlying architecture for the ReLu neural network? It would be a bit surprising if a feedforward neural network can capture the features from image inputs.

**Limitations:**

Yes

---

> ### Author Rebuttal · Authors · 2023-08-10
>
> Thank you for your time reviewing our paper, and for your insightful suggestions that have improved our work.
> > It would be more convincing if the authors can test on commonly used reinforcement learning benchmarks
>
> We conducted further experiments on two additional environments, Reacher-v4 and Cartpole-v1, from OpenAI Gym [3], and reported the results below (also in Rebuttal PDF Fig. 1)
> |                  |             Ours            |        MPPI       | Model-free PPO | PPO + Our model |     MBPO [4]    |
> |-------------|:-----------------------:|:---------------:|:--------------:|:---------------:|:---------------:|
> | Cartpole-v1 |  **0.003728** (4 ReLUs) | 0.003735 (full) |    0.039222    | 0.006291 (full) | 0.004268 (full) |
> |  Reacher-v4 | **0.053467** (48 ReLUs) | 0.064792 (full) |    0.237225    | 0.108016 (full) |  0.110031(full) |
>
> Each number represents the median of RMSE over 100 trials. Parentheses denote either the method used a full model or a sparsified model with specified number of ReLUs.
>
> As the table illustrates, our approach outperforms prior methods on the two standard RL benchmark environments. Notably, our approach achieved superior performance with highly sparsified neural dynamics models with fewer ReLUs compared to prior works.
>
> > The current results would leave the impression that the proposed method may overfit on these tasks only.
>
> Our experiments covered a variety of tasks, involving rigid and deformable objects in scenarios ranging from single to multi-objects, applied to dynamics models instantiated using feed-forward neural networks and graph-neural networks. These experiments demonstrated the applicability and combinatorial generalizability of our method. We hope that the results on the two additional tasks further substantiate that our method is applicable to a wide range of task settings.
>
> > What's the complexity of applying MIP?
>
> Solving MIPs is NP-hard, so in the worst case, the solve time might be exponential in the number of ReLUs. However, there exist highly optimized solvers like Gurobi that can solve medium-size MIPs very quickly in practice. The solution of a MIP can also be terminated early, and the branch and bound solver will return a solution together with an upper bound on the distance of this solution from the global minimum. This strategy can be leveraged to further reduce the runtimes, while still providing strong optimality guarantees. We provide details on various strategies in the literature to accelerate solving MIPs in Section 3.5.1 of the paper.
>
> > What's the underlying architecture for the ReLu neural network? It would be a bit surprising if a feedforward neural network can capture the features from image inputs.
>
> Below are the architecture details (also provided in supplemental material, section B). The ordered lists represent the number of ReLUs in each layer of the network.
>
> - PWA functions: 96, 192, 192, 96
>
> - Object pushing: 256, 256, 256
>
> - Object sorting: graph neural network similar to Sanchez-Gonzales et al. [5] with 64 hidden units per layer, with a total of 512 ReLUs
>
> - Rope manipulation: 256, 256, 256
>
> - Cartpole-v1: 16, 16, 16
>
> - Reacher-v4: 64, 64, 64
>
> We used either keypoint (object pushing, rope manipulation), object-centric representation (object sorting), or simulation state (cartpole, reacher) as our state representations. Prior works [1, 2] demonstrated that low-dimensional structured representations have superior generalization capabilities compared to pixel-based dynamics, especially for compositional systems containing varying numbers of objects (as is the case in our Object Sorting tasks).
>
> **Have these responses addressed the reviewer’s concerns? We look forward to continuing the discussion.**
>
> [1] N. Watters, A. Tacchetti, T. Weber, R. Pascanu, P. W. Battaglia, and D. Zoran, ‘Visual Interaction Networks’, CoRR, vol. abs/1706.01433, 2017.
>
> [2] D. Driess, Z. Huang, Y. Li, R. Tedrake, and M. Toussaint, ‘Learning Multi-Object Dynamics with Compositional Neural Radiance Fields’, in Conference on Robot Learning, 2022.
>
> [3] G. Brockman et al., ‘OpenAI Gym’, CoRR, vol. abs/1606.01540, 2016.
>
> ​​[4] M. Janner, J. Fu, M. Zhang, and S. Levine, ‘When to Trust Your Model: Model-Based Policy Optimization’, CoRR, vol. abs/1906.08253, 2019.
>
> [5] A. Sanchez-Gonzalez et al., ‘Graph networks as learnable physics engines for inference and control’, CoRR, vol. abs/1806.01242, 2018.

---

> > ### Author Response · Authors · 2023-08-18
> > **Are there any additional questions or concerns?**
> >
> > We hope that our additional experiments following the reviewer’s suggestions, discussion of limitations, and responses addressed the questions and concerns raised. **We would be happy to continue the discussion if the reviewer has additional questions or concerns.**

---

> > ### Comment · Reviewer_h4y2 · 2023-08-19
> >
> > Thanks to the authors for addressing my concern and questions. Regarding the additional experiments, could you also add (i) the accumulated rewards (ii) standard deviation, as in previous papers? After the authors' response, the strength of this submission is more clear to me. On the other hand, I also see the concern from the other reviewers on the experimental significance and extension to other complex domain, which seems a bit challenging given the current status.

---

> > > ### Author Response · Authors · 2023-08-19
> > >
> > > Thank you for the follow-up questions! We are glad to hear that our response addressed your concerns and questions. We respond to the additional questions below.
> > >
> > >
> > > > could you also add (i) the accumulated rewards (ii) standard deviation, as in previous papers?
> > >
> > > In Figure 1 of the Rebuttal PDF, we included the performance comparison of our approach against prior methods measured by RMSE, which in our setting is the additive inverse of the episode return averaged over steps. We included the interquartile range, minimum, and maximum as part of the box plot. These results further demonstrate that our approach achieved superior closed-loop control performance with sparsified dynamics models compared to prior works using full dynamics models.
> > >
> > > > I also see the concern from the other reviewers on the experimental significance and extension to other complex domain…
> > >
> > > We hope to highlight that **our approach achieved superior closed-loop control performance on the Object Sorting, Object Pushing, Cartpole, and Reacher tasks using highly sparsified neural dynamics models with fewer ReLUs, compared to model-free RL and prior methods using full dynamics models.**
> > >
> > > Only on the Rope Manipulation task, MIP (ours) achieves a similar performance to MPPI. The more significant gap in model prediction error between the full dynamics model and highly sparsified models limits the performance upper bound in closed-loop control, even though our approach using sparsified models enabled us to leverage a more powerful optimization tool giving better results in open-loop optimization. Future improvements could involve the co-optimization of the model sparsification procedure with control synthesis. Enabling the sparsification process to be mindful of the subsequent control task may lead to even better performance.
> > >
> > > Regarding the potential of extending our method to other complex domains, we would like to emphasize that **our method is generic and applicable to a wide variety of dynamics models instantiated using ReLU networks, including but not limited to feed-forward neural networks and graph-neural networks with compositional architectures.** Our experiments also showed **competitive performance and combinatorial generalizability on a wide range of tasks involving rigid and deformable objects in scenarios ranging from single to multiple objects.** The experiments on the two OpenAI Gym environments further demonstrated the generalizability of our method.
> > >
> > > **We wish to again express our heartfelt thanks for your helpful feedback and suggestions that helped us improve our paper. We would be happy to continue the discussion if the reviewer has additional questions or concerns.**

---

### Author Rebuttal · Authors · 2023-08-10

We thank the reviewers for dedicating their time and effort in reviewing our paper, and we deeply appreciate their thoughtful comments and insightful feedback. We appreciate the reviewers agreeing that our approach is novel, well formulated, tackles an interesting problem, and that our paper is well written.

> **Additional Experiments**

We have conducted additional experiments as requested by the reviewers and have included the results in the Rebuttal PDF, including:

1. Experiments involving two environments from OpenAI Gym (Cartpole-v1, Reacher-v4), to further demonstrate the generalizability of our method to a variety of tasks. (Reviewer h4y2)
2. Comparisons to an additional prior model-based RL method (MBPO), to showcase the superior performance of our method using a sparsified dynamics model compared to prior methods using a full dynamics model. (Reviewers H7kE, LeuW)
3. Taking dynamics models trained and sparsified through our method to train a control policy with PPO, demonstrating that our sparsification scheme produces dynamics models that are general and can be effectively combined with prior model-based RL methods. (Reviewers H7kE, LeuW)
4. Closed-loop control performance of dynamics models sparsified using our approach compared to models sparsified using a prior method, showing that our sparsification approach generates dynamics models with lower prediction errors and better closed-loop control performance. (Reviewer fLUK)
5. Experiments examining the positive correlation between computation complexity and the number of remaining ReLUs in the dynamics model, to further support using remaining ReLUs as an effective proxy for computation complexity. (Reviewers h4y2, LeuW)

> **Applicability of the Proposed Method**

Following the comments from Reviewers h4y2, H7kE, and LeuW about the applicability of our method, we hope to highlight that our method is generic and applicable to a wide variety of dynamics models instantiated using ReLU networks including but not limited to feed-forward neural networks and graph-neural networks examined in this paper. Our experiments also showed competitive performance and combinatorial generalizability on a wide range of tasks involving rigid and deformable objects in scenarios ranging from single to multi-objects.


> **Limitations**

In response to reviewers’ requests, we provide more details on the limitations of our approach below. We will include these discussions in the revised version of our paper.

Our method relies on sparsifying neural dynamics models to fewer ReLU units to make the control optimization process solvable in a reasonable time due to the worst-case exponential run time of MIP solvers. Although our experiments showed that this already enabled us to complete a wide variety of tasks, our approach may struggle when facing a much larger neural dynamics model. We provided a review of a variety of strategies in the literature to accelerate the process of finding a solution for our MIPs in section 3.5.1 of the paper.

Our experiments also demonstrated superior closed-loop control performance using sparsified dynamics models with only reasonably good prediction accuracy as a result of benefiting from stronger optimization tools, but our approach may suffer if the sparsified dynamics model becomes significantly worse and incapable of providing useful forward predictions.

In terms of generality, our method naturally applies to dynamics models represented by ReLU networks with diverse model architectures (MLP, GNN, etc.), and showed superior performance on a variety of tasks involving rigid and deformable objects in single to multi-object scenarios, demonstrating the applicability and combinatorial generalizability of our method.

**We look forward to discussing further with the reviewers during the discussion phase.**

---

### Decision · Program_Chairs · 2023-09-21

**Decision:**

Accept (poster)

**Comment:**

Thank you for your submission and active engagement throughout the review period. After the discussion period, the reviewers and I are in agreement that the paper will be a great addition to the conference. The idea of learning/pruning for sparse dynamics that can be used with a MIP planner is appealing and the experiments in comparison to MPPI and other baselines are well-executed. Two concerns came up during the discussion period that we encourage the authors to incorporate into the final version of the paper:

1.  On the ability of obtaining a globally optimal solution, the paper states "if only a few ReLUs are left in the model, Equation 10 can be efficiently solved to global optimality." However, depending on the cost function, even if the dynamics are formed as MIP, global optimality is not easy. Restricting the model flexibility to enable efficient planning is not new. For example, one can also exploit deep Koopman operator [1], whose planning procedure can be completed via LP, instead of MIP in this paper. However, this has not been discussed.
2. While the experimental demonstrations on the robotic environments are compelling, the fact that they are non-standard makes it difficult to understand if the method would work in other control and model-based reinforcement learning settings where the dynamics models are learned.

[1] Learning Compositional Koopman Operators for Model-Based Control. Yunzhu Li, Hao He, Jiajun Wu, Dina Katabi, Antonio Torralba